# Investigating the Pulmonary Host Response of *Acinetobacter baumannii* Infection-Associated Pneumonia by Metagenomic Next-Generation Sequencing

**DOI:** 10.3390/biomedicines13010142

**Published:** 2025-01-09

**Authors:** Mu-Jung Chou, Chih-Hung Cheng, Hui-Ching Wang, Ming-Ju Tsai, Chau-Chyun Sheu, Wei-An Chang

**Affiliations:** 1Department of Internal Medicine, Kaohsiung Medical University Hospital, Kaohsiung Medical University, Kaohsiung 807, Taiwan; fatty8264@gmail.com (M.-J.C.); markbruse617@gmail.com (C.-H.C.); siegfriedtsai@gmail.com (M.-J.T.); sheucc@gmail.com (C.-C.S.); 2Division of Pulmonary and Critical Care Medicine, Kaohsiung Medical University Hospital, Kaohsiung 807, Taiwan; 3Department of Internal Medicine, School of Medicine, College of Medicine, Kaohsiung Medical University, Kaohsiung 807, Taiwan; 4Department of Nursing, Kaohsiung Medical University Hospital, Kaohsiung Medical University, Kaohsiung 807, Taiwan; 830126@kmuh.org.tw; 5Graduate Institute of Clinical Medicine, College of Medicine, Kaohsiung Medical University, Kaohsiung 807, Taiwan

**Keywords:** mNGS, *Acinetobacter baumannii*, NDM, immune, pulmonary host response, ferroptosis

## Abstract

**Background:** For investigating the host response in *Acinetobacter baumannii* associated pneumonia, we analyzed the host genetic sequences obtained from metagenomic next-generation sequencing (mNGS). **Methods:** The samples for mNGS were bronchoalveolar lavage fluid (BALF) collected from the lungs of patients infected with *A. baumannii* and from patients without bacterial infections. BALF samples from patients with pneumonia were collected from the lungs of patients infected with *A. baumannii* with New Delhi metallo-β-lactamase (NDM, before treatment), A. baumannii with NDM (post-treatment), *A. baumannii* without resistant genes, and those without bacterial infection. Partek was used for investigating enriched functions and pathways related to the pulmonary host response to pneumonia caused by *A. baumannii* with NDM infection and *A. baumannii* without antimicrobial-resistant genes. The STRING was employed for identifying protein interaction pathways related to the pulmonary host response to pneumonia caused by *A. baumannii* without antimicrobial-resistant genes. **Results:** In pulmonary host response to pneumonia caused by *A. baumannii* with NDM, five immune system-related pathways and five pathways related to signal transduction were identified. No significant differences were observed in the immune system and signal transduction pathways in the pulmonary host response to pneumonia caused by *A. baumannii* without antimicrobial-resistant genes. However, significant differences were noted in the phagosome, ferroptosis, and regulation of the actin cytoskeleton in cellular processes. **Conclusions:** mNGS provides information not only on pathogen gene expression but also on host gene expression. In this study, we found that pneumonia with *A. baumannii* carrying the NDM resistance gene triggers stronger immune responses in the lung, while pneumonia with *A. baumannii* lacking antimicrobial resistance genes is more linked to iron-related pathways.

## 1. Introduction

Hospital-acquired bacterial pneumonia (HABP) and ventilator-associated bacterial pneumonia (VABP) are healthcare-associated infections associated with mortality, high costs, and antibacterial drug resistance [1]. Significantly, the frequency of multidrug-resistant (MDR) gram-negative (GN) bacteria as the bacteriologic cause of HABP/VABP is increasing [2]. These bacteria include MDR *Pseudomonas aeruginosa* (*P*. *aeruginosa*), *Acinetobacter baumannii* (*A*. *baumannii*), and carbapenem-resistant Enterobacteriaceae [2]. Acinetobacter spp. are glucose-non-fermentative, nonmotile, non-fastidious, catalase-positive, oxidative-negative, and aerobic GN coccobacilli [3]. Several reports have shown that *A*. *baumannii* rapidly develops resistance to antimicrobials through various mechanisms, including enzymatic degradation, target modifications, multidrug efflux pumps, and permeability defects, thereby leading to the isolation of MDR strains [4]. NDM (New Delhi metallo-β-lactamase) enzymes, such as NDM-1, NDM-2, NDM-3, NDM-5, and NDM-7, have been detected in *A. baumannii*, contributing to its carbapenem resistance. The blaNDM-1 gene is often associated with other resistance determinants, resulting in multidrug resistance. This makes *A. baumannii* infections extremely difficult to treat [5].

Moreover, *A. baumannii* employs phospholipase and metal acquisition mechanisms. It uses phospholipase to disrupt the host cell membranes to obtain essential nutrients. In addition, it can acquire metals, such as iron and zinc, from host cells through various iron acquisition systems (e.g., siderophores), enhancing its survival capabilities. These mechanisms enable *A*. *baumannii* to withstand challenging hospital environments, resist multiple antibiotics, and pose significant challenges in clinical treatment [6].

Recent developments in omics-based technologies and system medicine offer the potential for addressing these challenges by delivering in-depth insights into immune endotypes. This stratification of patients enables the implementation of precision medicine approaches that tailor immunotherapies for sepsis according to specific biomarkers and molecular mechanisms [7].

Metagenomic next-generation sequencing (mNGS) is an advanced method that integrates high-throughput sequencing with bioinformatics analysis, enabling the rapid and accurate detection of various pathogens, including bacteria, fungi, viruses, and parasites, simultaneously from DNA or RNA gene sequencing of clinical samples [8].

Both the pathogen and host genetic sequences are sequenced during mNGS. In clinical applications, mNGS is used to detect pathogen genes, so human genes are not utilized. [9]. In this study, we utilized the host genetic sequences, which would normally be discarded, for the bioinformatics analysis. The samples used were bronchoalveolar lavage fluid (BALF) collected from the lungs of patients infected with *A*. *baumannii* and from patients without bacterial infections. Through this approach, we aimed to explore the host response generated by *A*. *baumannii* in the lungs of patients with HABP.

## 2. Materials and Methods

### 2.1. Flowchart of the Study

The flowchart of the study design is illustrated in Figure 1. BALF samples from patients with pneumonia were collected from the lungs of patients infected with *A. baumannii* with New Delhi metallo-β-lactamase (NDM, before treatment), *A. baumannii* with NDM (post-treatment), *A. baumannii* without antimicrobial-resistant genes, and those without bacterial infection. After conducting mNGS, the host RNA portion was extracted for analysis. This study used two bioinformatics tools, Partek and STRING. Partek was used for investigating enriched functions and pathways related to the pulmonary host response to pneumonia caused by *A. baumannii* with NDM and *A. baumannii* without antimicrobial-resistant genes. The STRING was employed for identifying protein interaction pathways related to the pulmonary host response to pneumonia caused by *A. baumannii* without resistant genes. RNA data collected after treatment for pneumonia caused by *A. baumannii* with NDM were used only for normalization and were excluded from the pathway analysis.

### 2.2. Patients’ Clinical Data and Results of the Pathogens

We used BALF samples from three patients admitted to the medical intensive care unit (ICU) due to pneumonia and acute respiratory failure for mNGS. As all three patients underwent endotracheal intubation and were on mechanical ventilation, the BALF samples were collected via bronchoscopy through the endotracheal tube at the suspected pneumonia sites. Before collection, physicians determined the site for BALF collection on the basis of the patients’ chest X-ray images. As shown in Table 1, the mNGS results for Patient 1 do not show bacterial infection; however, a *Candida albicans* (*C. albicans*) infection may have existed. The BALF sample of Patient 2 showed a predominant *A. baumannii* infection, with a reads count of 14,557, and the film array results indicated the presence of a resistant gene, New Delhi metallo-β-lactamase (NDM). The BALF sample of Patient 3 revealed a predominant *A. baumannii* infection, with a reads count of 140,867, and the film array results showed no resistant genes. Based on these findings, we defined Patient 1 as having no bacterial infection, Patient 2 as *A. baumannii* with NDM, and Patient 3 as *A. baumannii* without resistant genes (Table 1).

### 2.3. mNGS Process

Following sample collection, they were immediately stored in a freezer with a temperature of 80 °C. Meanwhile, the BALF samples were performed with the mNGS method for DNA and RNA tests in parallel (Asia Pathogenomics Co., Ltd., New Taipei City, Taiwan).

#### 2.3.1. Nucleic Acid Extraction

All BALF samples were heat-inactivated at 65 °C for 30 min. For genomic DNA (gDNA) extraction, 600 μL of the inactivated BALF samples were transferred into a 2 mL Lysing Matrix tube containing 1 g of 0.5 mm diameter glass beads. The tube was placed on a FastPrep-24 5G instrument (MP Biomedicals, Santa Ana, CA, USA) at a speed of 10 m/s for 25 min. Subsequently, the supernatant (200 μL) was transferred to a new 1.5 mL tube following high-speed centrifugation at 8000× *g* for 1 min for further DNA extraction. gDNA extraction was performed using the TIANMicrobe Pathogen DNA Kit (Tiangen Biotech, Beijing, China) following the manufacturer’s instructions. The total gDNA was quantified using the Qubit dsDNA HS Assay Kit on a Qubit 4.0 Fluorometer (Thermo Fisher Scientific. Waltham, MA, USA) [10].

For total RNA extraction, 150 μL of the inactivated BALF samples were transferred into a 1.5 mL tube for host gDNA removal by low-speed centrifugation at 1500× *g* for 30 min. The supernatant (140 μL) was subsequently transferred to a new 1.5 mL tube for further RNA extraction. Total RNA extraction was performed using the QIAamp Viral RNA Mini Kit (Qiagen, Hilden, Germany) following the manufacturer’s instructions. The total RNA was quantified using the Qubit RNA HS Assay Kit on a Qubit 4.0 Fluorometer (Thermo Fisher Scientific) [11,12].

#### 2.3.2. DNA Library Preparation

Approximately 100 ng of input gDNA was used for DNA library construction, following the MGIEasy FS DNA Library Prep Kit protocol (MGI, Shenzhen, China). The procedure included DNA fragmentation, end-repair, adapter ligation, and polymerase chain reaction (PCR) amplification. In brief, DNA fragmentation was performed using an enzymatic method with a fragmentation enzyme to generate DNA fragments of approximately 150–250 bp. This procedure was achieved by incubating the sample at 32 °C for 20 min. The fragmented products were purified using 0.8× DNA purification beads, followed by two washes with fresh 80% ethanol, and eluted in a 42 µL TE buffer. The purified DNA fragments were end-repaired and A-tailed under the following conditions: 37 °C for 10 min, followed by 65 °C for 15 min. Subsequently, adapters were ligated to the end-repaired and A-tailed DNA fragments at 23 °C for 20 min, followed by bead purification at a 0.5× ratio. A unique barcode was assigned to each library.

Library amplification was performed using KAPA HiFi polymerase (Roche, Wilmington, MA, USA) under the following PCR conditions: 95 °C for 3 min; 12 cycles of 98 °C, 60 °C, and 72 °C for 20, 15, and 30 s, respectively; followed by a final extension at 72 °C for 10 min; and held at 4 °C. The amplified PCR product was purified using 0.8× bead purification [10,13].

#### 2.3.3. RNA Library Preparation

RNA library preparation was performed using the MGIEasy RNA Library Prep Kit (MGI, Shenzhen, China) following the steps of fragmentation, reverse transcription, second-strand synthesis, end-repair, adapter ligation, and PCR amplification. In brief, the total RNA was converted to complementary DNA (cDNA) during the initial three steps. The cDNA products were purified using 1.5× purification beads, followed by two washes with fresh 80% ethanol, and eluted in a 42 µL TE buffer. The remaining library construction steps were identical to those used for DNA library preparation, except for the PCR conditions, which were as follows: 95 °C for 3 min; 14 cycles of 95 °C, 56 °C, and 72 °C for 30, 30, and 60 s, respectively; followed by a final extension at 72 °C for 5 min; and held at 4 °C. The PCR product was purified using 1.2× bead purification. The quality of all sequencing libraries was assessed using a Qubit 4.0 Fluorometer (Thermo Fisher Scientific) [12].

#### 2.3.4. Library Pooling and DNA Nanoball (DNB) Sequencing

The DNA or RNA library was allocated to 40 million reads. Subsequently, to generate single-stranded DNA, the pooled DNA library was denatured at 95 °C for 6 min, followed by circularization using ligase at 37 °C for 30 min. The resulting single-stranded circularized DNA library was transformed into DNBs using DNB polymerase I, provided by the DNBSEQ-G50RS FCL SE50 sequencing kit (MGI, Shenzhen, China), at 30 °C for 25 min. Sequencing was performed on the DNBSEQ-G50RS platform using the DNBSEQ-G50RS sequencing flow cell [10,13,14].

### 2.4. Partek

The computational platform of the Partek Flow^®^ software (version 12.0.1) was used for RNA quantification and identification of significant pathways for the differentially expressed genes according to the Kyoto Encyclopedia of Genes and Genomes (KEGG) database. RNA reads were normalized using the median ratio (DESeq2 only) method. The pathways with a *p*-value of <0.05 were selected as statistically significant candidates [15].

### 2.5. STRING Database Analysis

Functional interactions between proteins play a crucial and complex role within cells. The STRING database (https://string-db.org/, accessed on 26 November 2024) aggregates and integrates these data by compiling known and predicted protein–protein interactions for a wide range of organisms. STRING provides information on both direct (physical) and indirect (functional) interactions. These associations were derived from the following five primary sources: conserved co-expression patterns, high-throughput laboratory experiments, automated text mining, genomic context predictions, and previously established knowledge within the database [16].

## 3. Results

### 3.1. Information on the Pathogens and Clinical Data of Patients

We used BALF samples from three patients admitted to the medical ICU due to pneumonia and acute respiratory failure for mNGS analysis. As all three patients had undergone endotracheal intubation and were on ventilators, the BALF samples were collected using a bronchoscope through the endotracheal tube, targeting the suspected pneumonia site. Before sample collection, physicians determined the BALF collection site on the basis of the patient’s chest X-ray images. The mNGS results of the BALF sample of Patient 1 showed no bacterial infection but indicated a possible *C*. *albicans* infection (Table 1). The mNGS results of the BALF sample of Patient 2 primarily indicated an *A*. *baumannii* infection, with a read count of 14,557. The film array results further revealed the presence of the resistant gene. NDM. In the BALF sample of Patient 3, the mNGS results also showed a predominant *A*. *baumannii* infection, with a reads count of 140,867; however, the film array results showed no resistant genes. Based on these findings, we classified Patients 1, 2, and 3 as “no bacterial infection,” “*A*. *baumannii* with NDM infection”, and “*A*. *baumannii* without resistant gene infection”, respectively.

### 3.2. Differential Gene Expressions in A. baumannii with NDM and A. baumannii Without Resistant Gene Infection-Associated Pulmonary Host Response

The volcano plots of the differentially expressed genes between nonbacterial infection and *A*. *baumannii* with NDM infection in the BALF samples are depicted in Figure 2a. The volcano plots of the differentially expressed genes between nonbacterial infection and *A*. *baumannii* infection in the BALF samples are displayed in Figure 2b. The colored plots indicate the significantly dysregulated genes (fold change > 2; *p* < 0.05). A total of 2211 significantly upregulated genes and 2296 significantly downregulated genes were identified in *A*. *baumannii* with NDM host response, and a total of 2873 significantly upregulated genes and 3098 significantly downregulated genes were identified in *A*. *baumannii* without resistant gene host response.

### 3.3. KEGG Pathways Associated with the Immune System and Cellular Processes

The KEGG pathways in *A*. *baumannii* with NDM and *A*. *baumannii* pulmonary host response were predicted using the Partek software (build version 12.0.1). The differentially expressed genes in *A*. *baumannii* with NDM and *A*. *baumannii* pulmonary host responses were analyzed. In the *A*. *baumannii* with NDM group, 344 KEGG pathways were identified, with 51 pathways having a *p*-value of <0.05. In the *A*. *baumannii* group, 329 KEGG pathways were identified, with 18 pathways having a *p*-value of <0.05. The 10 most significant pathways from both groups are listed in Table 2.

We compiled the relevant KEGG pathways to explore the expression of the immune system, signal transduction, and cellular processes in *A*. *baumannii* with NDM and *A*. *baumannii* without resistant gene infection-related pulmonary host response (Table 3). In the *A*. *baumannii* with NDM pulmonary host response, the following five immune system-related pathways were noted: the thyroid hormone signaling pathway, parathyroid hormone synthesis, glucagon signaling pathway, insulin signaling pathway, and cortisol synthesis and secretion. The following four pathways related to signal transduction were identified: endocytosis, the ErbB signaling pathway, the AMPK signaling pathway, and actin cytoskeleton regulation. The HIF-1 signaling pathway was classified under signaling molecules and interaction. No significant differences were observed in the immune system and signal transduction pathways in the *A*. *baumannii* without resistant gene pulmonary host response. However, significant differences were noted in the phagosome, ferroptosis, and regulation of the actin cytoskeleton in cellular processes.

### 3.4. Iron-Related Pathways in A. baumannii Without Resistant Gene Infection Pulmonary Host Response

In the significantly expressed KEGG pathways in *A*. *baumannii* without resistant gene infection-associated pulmonary host response, 57 differentially expressed genes were involved in the cellular process pathways. To understand the role of iron-related pathways in the *A*. *baumannii* without resistant gene pulmonary infection host response, we employed the STRING to analyze these 57 genes. In the STRING, each gene was considered a node, and these 57 nodes generated 199 edges (Figure 3).

Here, we used k-means clustering to divide these 57 genes into the following 3 clusters: red, related to the regulation of the actin cytoskeleton and EPH–Ephrin signaling; blue, related to the antigen processing and presentation of exogenous peptide antigens; and green, related to ferroptosis and iron uptake. Regarding ferroptosis and iron uptake, the STRING provided the pathways and functions that may be influenced, with the iron-related parts being shown in Figure 4. The pathway-related genes are listed next to the respective pathway, with red indicating increased expression and black indicating decreased expression.

## 4. Discussion

Nosocomial pneumonia can have a crude mortality rate reaching 70%, with *P. aeruginosa* and *Acinetobacter* spp. infections causing higher death rates. A significant proportion of deaths related to ventilator-associated pneumonia (VAP) are directly caused by the infection; however, recent studies have suggested that the attributable mortality of VAP is approximately 10%. Surgical patients and those with moderate illness severity are at the highest risk, with the impact of VAP being influenced by time-dependent factors, such as ICU discharge and overall mortality [17]. New Delhi metallo-β-lactamase (NDM) is a powerful enzyme that hydrolyzes most β-lactams, with variants identified in over 60 species of bacteria, predominantly in Klebsiella pneumoniae and Escherichia coli. NDM-positive strains are found worldwide, particularly in the Indian subcontinent, the Middle East, and the Balkans. Despite the detection of compounds to inhibit NDM, no clinical treatments are approved yet, and outbreaks continue to pose a significant healthcare challenge, requiring further research on prevention and control measures [18]. This study aims to investigate the role and impact of NDM in the pneumonia microenvironment caused by *A. baumannii*, in order to elucidate its potential effects on the progression of infection.

*A. baumannii* is a GN opportunistic pathogen that causes severe hard-to-treat infections, with increasing infection rates and a major role in global nosocomial infections. Its abilities to form biofilms, adapt to various environments, and exhibit multidrug resistance, including carbapenem resistance, make it a significant concern in both human and veterinary medicine [19].

Sepsis remains a leading cause of illness and death worldwide, and although antibiotic therapy and supportive care have improved survival, further advancements have been difficult owing to the complexity of immune responses in patients. Recent developments in omics-based technologies and system medicine offer hope for more precise biomarker-driven immunotherapy approaches that could tailor treatment to specific immune endotypes, potentially making immunotherapy a key part of future sepsis treatment [7].

Therefore, the immune response plays a crucial role in the context of infectious diseases. However, detecting immune-related markers in samples or conducting whole-genome sequencing requires substantial costs. We here employed mNGS to identify the pathogens in the samples. Furthermore, bioinformatic analysis was performed on the host sequences produced by mNGS, which were initially unused. We cross-referenced the information and discovered new pathogenic mechanisms by combining clinical data from patients with the pathogen types identified using mNGS, as well as the host sequences processed through bioinformatic analysis. In our study, the *A. baumannii* with NDM group had more immune-related pathways than the *A. baumannii* without resistant gene group. The thyroid hormone signaling pathway, parathyroid hormone synthesis, glucagon signaling pathway, insulin signaling pathway, and cortisol synthesis and secretion were involved in the pulmonary infection host response of *A. baumannii* with NDM. These pathways may serve as potential therapeutic targets.

The *A. baumannii* without resistant gene group had more iron-related pathways, including ferroptosis, which is an iron-dependent form of cell death regulated by autophagy, specifically through iron storage protein degradation via ferritinophagy. This process, mediated by the cargo receptor NCOA4, supports ferroptosis by maintaining cellular iron balance and promoting reactive oxygen species accumulation [20].

A defense mechanism called nutritional immunity is used by the human body to limit the availability of extracellular iron, thereby restricting pathogens from accessing this vital metal. By sequestering iron and contributing to the body’s immune defense, proteins, including transferrin, lactoferrin, ferritin, and the innate immune protein calprotectin, which binds iron and other essential elements, play key roles in this process [21]. Human transferrin inhibited the growth of *A. baumannii* by sequestering iron and disrupting their membrane potential. No resistance emerged even after repeated exposure to subtherapeutic levels of transferrin. Intravenous human transferrin-treated infected mice showed improved survival and reduced microbial load [22].

Ferritin heavy chain 1 (FTH1), ferritin light chain (FTL), mitochondrial ferritin (FTMT), and ATP6V1F were the downregulated genes identified in *A. baumannii* without resistant gene infection-associated pulmonary host response. FTH1 is crucial for maintaining cellular iron homeostasis during ferroptosis. Notably, FTH1 is also implicated in ferritinophagy, a specific type of autophagy. Ferritinophagy involves the targeted breakdown of ferritin through autophagy, supplying the necessary substrates that contribute to ferroptosis [23]. To increase free iron levels, ferritin, composed of FTL and FTH1, can be degraded by lysosomes, and inhibiting NCOA4-mediated ferritinophagy enhances iron storage while reducing ferroptosis in cancer cells. Moreover, the overexpression of FTMT and the iron chaperone activity of PCBP1, which delivers Fe2+ to ferritin, both contribute to limiting ferroptosis in various cell types [24]. FTMT, which is structurally similar to cytosolic FTH1, plays a protective role in erastin-induced ferroptosis by blocking the increase in the labile iron pool and reactive oxygen species, thereby leading to prolonged survival in a Drosophila ferroptosis model [25]. ATP6V1F encodes a component of vacuolar ATPase, which is crucial for organelle acidification and plays a role in various processes, including zymogen activation, endocytosis, and protein sorting. ATP6V1F is overexpressed in hepatocellular carcinoma (HCC), and this overexpression is associated with poor prognosis, immune cell infiltration, and elevated expression of immune checkpoints [26].

Fibroblast growth factor (FGF)-23, TFRC, FGFR4, and ATPase H+ transporting V0 subunit B (ATP6V0B) were the upregulated genes identified in *A. baumannii* without resistant gene infection-associated pulmonary host responses. FGF23 is a key regulator of phosphate and calcium homeostasis and is also linked to iron metabolism, inflammation, and erythropoiesis. New techniques using liquid bone biopsy to assess FGF23 dynamics can infer its transcription, post-translational modification, and peptide cleavage in bone, advancing research on human diseases. Understanding the relationship between FGF23, iron homeostasis, and erythropoiesis may lead to novel therapies for anemia and FGF23 excess conditions, such as chronic kidney disease [27]. The TFRC gene encodes the transferrin receptor protein 1 (TFR1) in humans, which regulates iron uptake into cells from the extracellular environment. TFR1 plays a vital role in facilitating ferroptosis by maintaining intracellular iron levels [28]. FGFR4 enhances metastasis, angiogenesis, chemoresistance, and cancer cell stemness across various digestive system tumors [29]. Lenvatinib induces ferroptosis by inhibiting FGFR4, and the involvement of Nrf2 influences the sensitivity of HCC to this treatment [30]. The ATP6V0B gene plays a crucial role in regulating the immune response to Vibrio vulnificus infection in half-smooth tongue sole, with its expression significantly upregulated in key tissues, including the liver, spleen, blood, and heart tissues, during the early stages of infection [31]. These ferroptosis-related genes may play a crucial role in the host response to *A. baumannii* infection-associated pulmonary infection and could potentially serve as new therapeutic targets in the future. The study had several limitations. First, the sample size was small, with only four samples analyzed. Second, the mNGS results for the three patients detected other potential pathogens, indicating that the infections were not exclusively caused by *A. baumannii*. Lastly, no protein expression levels were measured to validate the findings. A larger sample size and more relevant clinical characteristics for validation are needed.

From the above results, it can be observed that the molecular mechanisms caused by *A. baumannii* with antibiotic resistance genes and those without antibiotic resistance genes are different in the pneumonia lung microenvironment. *A. baumannii* with resistance genes seem to produce a more diverse set of molecular mechanisms. This may suggest that the impact of antibiotic resistance genes is not limited to just antibiotic resistance, but also includes other pathogenic mechanisms (Figure 5).

## 5. Conclusions

In this study, we found that pulmonary host responses to pneumonia caused by *A. baumannii* with the NDM resistance gene involved significantly more prominent immune responses. On the other hand, host responses to pneumonia caused by *A. baumannii* lacking antimicrobial resistance genes were more closely associated with iron-related pathways. These findings may be beneficial for the development of future treatments for pneumonia (Figure 5).

## Figures and Tables

**Figure 1 biomedicines-13-00142-f001:**
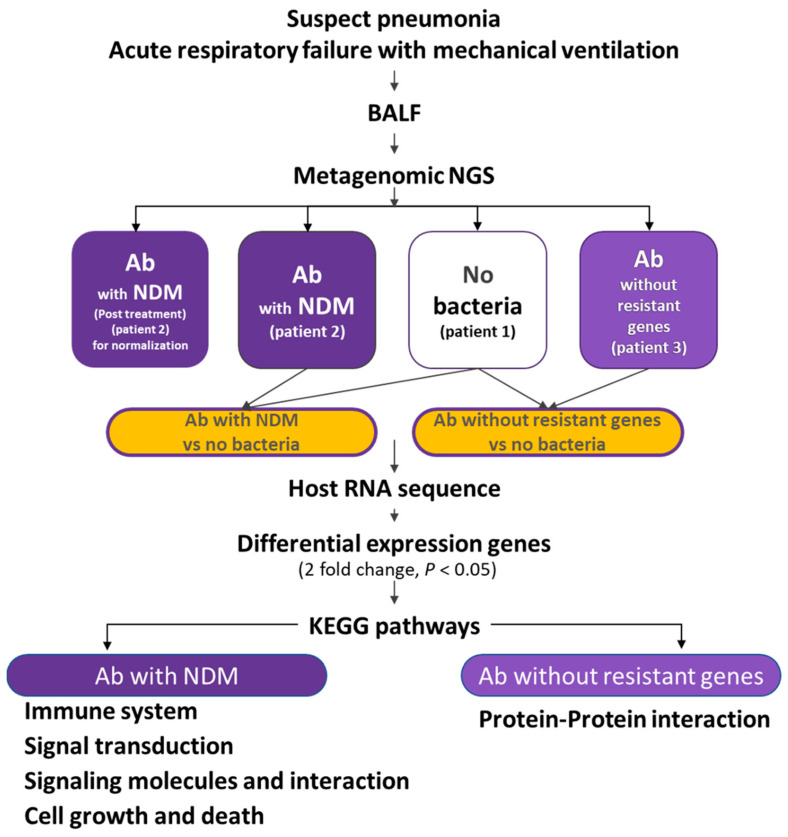
Study flowchart. BALF samples were collected from patients with pneumonia, including those infected with *A. baumannii* NDM (pre- and post-treatment), *A. baumannii* without antimicrobial-resistant genes, and patients without bacterial infections. After mNGS, the host RNA was analyzed using two bioinformatics tools: Partek and STRING. Partek was used to explore enriched functions and pathways linked to the pulmonary host response to pneumonia caused by *A. baumannii* with and without resistant genes. STRING focused on identifying protein interaction pathways related to the host response to pneumonia caused by *A. baumannii* without resistant genes. RNA data collected after treatment for pneumonia caused by *A. baumannii* with NDM were used only for normalization and excluded from pathway analysis. NGS: next-generation sequencing, BALF: bronchoalveolar lavage fluid, Ab: *Acinetobacter baumannii*, NDM: New Delhi metallo-β-lactamase, KEGG: Kyoto Encyclopedia of Genes and Genomes.

**Figure 2 biomedicines-13-00142-f002:**
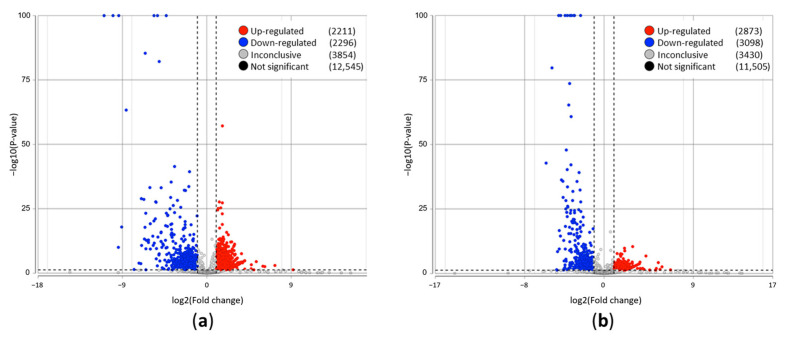
Overview of the gene expression profile in *Acinetobacter baumannii* (*A*. *baumannii*) infection-associated pulmonary host response. (**a**) Volcano plot of differential gene expression patterns of the bronchoalveolar lavage fluid (BALF) sample from the *A*. *baumannii* with NDM-infected lung versus the BALF sample from the lung without bacterial infection. The significantly dysregulated genes in the BALF sample from the *A*. *baumannii*-infected lung (those with −log10 [*p*-value] > 1.3; fold change > 2) are shown in blue (downregulation) and red (upregulation). (**b**) Volcano plot of differential gene expression patterns of the BALF sample from the *A*. *baumannii*-infected lung versus the BALF sample from the lung without bacterial infection. The significantly dysregulated genes in the BALF sample from the lung with *A*. *baumannii* without resistant gene infection (those with −log10 [*p*-value] > 1.3; fold change > 2) are shown in blue (downregulation) and red (upregulation).

**Figure 3 biomedicines-13-00142-f003:**
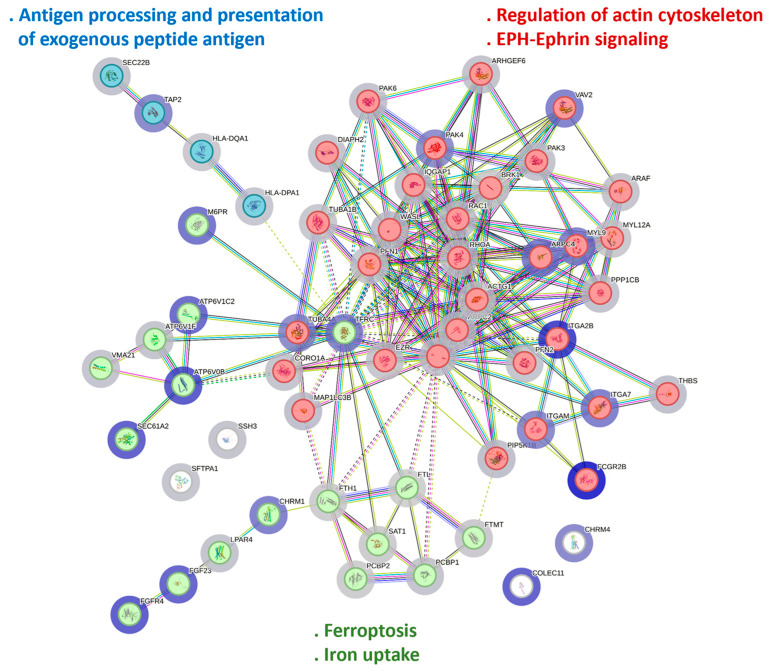
Protein–protein interaction network analysis in *A*. *baumannii* without resistant gene infection-associated pulmonary host response. Fifty-seven differentially expressed genes involved in cellular process pathways are used to investigate the role of iron-related pathways in the *A*. *baumannii* without resistant gene infection pulmonary host response. In the STRING, each gene is considered a node, and these 57 nodes generated 199 edges. The minimum required interaction score is set to medium confidence (score = 0.400). This analysis obtained a highly interactive PPI network of 57 nodes and 199 edges, with a PPI enrichment *p*-value of <1.0 × 10^−16^. We used k-means clustering to divide these 57 genes into the following three clusters: red, related to the regulation of the actin cytoskeleton and EPH–Ephrin signaling; blue, related to the antigen processing and presentation of exogenous peptide antigens; and green, related to ferroptosis and iron uptake.

**Figure 4 biomedicines-13-00142-f004:**
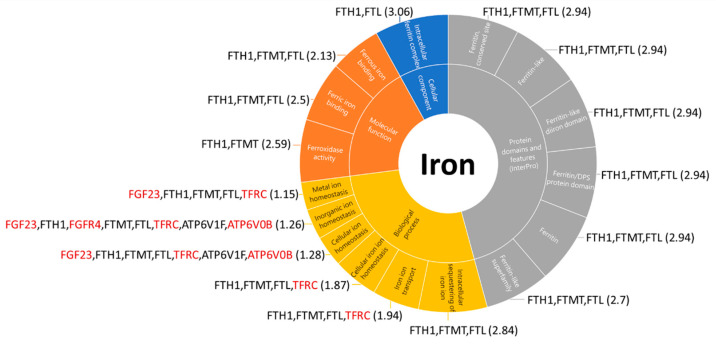
Iron-related pathways and dysregulated genes in *A*. *baumannii* without resistant gene infection-associated pulmonary host response. Regarding ferroptosis and iron uptake, the STRING provides the pathways and functions that may be influenced, by the iron-related parts. The genes related to the pathway will be listed next to the respective pathway, with red indicating increased expression and black indicating decreased expression. All of these pathways have an FDR of <0.05.

**Figure 5 biomedicines-13-00142-f005:**
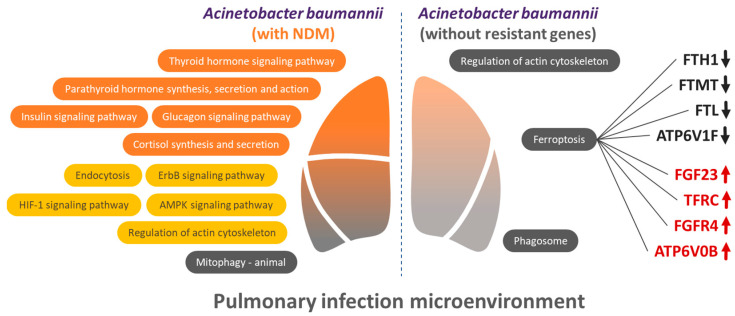
Schematic summary of the molecular mechanisms potentially involved in the *A*. *baumannii* infection-associated pulmonary host response.

**Table 1 biomedicines-13-00142-t001:** Information on the pathogens and clinical data of patients.

	Patient 1	Patient 2	Patient 3
Sex	F	M	M
Age (years)	79	78	72
ICU length of stay (days)	29	32	22
Cause of ICU admission	Pneumonia, acute respiratory failure	Pneumonia, acute respiratory failure	Pneumonia, acute respiratory failure
SOFA on BALF (days)	11	9	6
Mortality	Y	Y	Y
Cancer	N	N	N
COVID-19	N	Y	N
DM	N	N	N
Hypertension	N	Y	N
Other underlying diseases	Sicca syndrome, dermatomyositis, dementia	None	Dementia, traumatic brain injury, atrial fibrillation
BALF culture report	None	*Acinetobacter baumannii* *Candida albicans*	None
Film array report	None	*Acineto calc baumannii* complex (10^7^ copy/mL) *Klebsiella pneumoniae* group (10^5^ copy/mL)	*Acineto calc baumannii* complex (10^6^ copy/mL) No resistant genes
Resistant genes	None	NDM	none
mNGS report	*Candida albicans* (15,740) Human γ-gamma herpes virus 4 (13)	*Acinetobacter baumannii* (14,557) *Klebsiella pneumoniae* (4) *Chryseobacterium indologenes* (4) *Candida albicans* (585) Human α-herpes virus 1 (6) COVID-19 (55)	*Acinetobacter baumannii* (140,867) *Corynebacterium striatum* (39,157) *Chryseobacterium indologenes* (21,651) *Candida albicans* (15) Human β-herpes virus 7 (25) Human α-herpes virus 1 (11) Human β-herpes virus 5 (6)

ICU, intensive care unit; SOFA, sequential organ failure assessment; BALF, bronchoalveolar lavage fluid; COVID-19, coronavirus disease 2019; DM, diabetes mellitus; mNGS, metagenomic next-generation sequencing; NDM, New Delhi metallo-β-lactamase. In the mNGS report section, the numbers in parentheses refer to the reads number obtained from mNGS.

**Table 2 biomedicines-13-00142-t002:** Top 10 KEGG pathways involved in the *Acinetobacter baumannii* with NDM infection-associated pulmonary host response.

Group	Enrichment Score	*p*-Value	Genes in the List
*Acinetobacter baumannii* with NDM vs. nonbacterial infection			
Ribosome	32.13	0.00	72
COVID-19	27.20	0.00	91
Thyroid hormone signaling pathway	9.78	0.00	42
Primary immunodeficiency	5.96	0.00	15
Glycosphingolipid biosynthesis: lacto and neolecto series	5.89	0.00	12
Gastric cancer	5.61	0.00	43
Arrhythmogenic right ventricular cardiomyopathy	5.35	0.00	25
Nicotine addiction	5.09	0.01	15
Vasopressin-regulated water reabsorption	5.00	0.01	16
Herpes simplex virus 1 infection	4.92	0.01	118
*Acinetobacter baumannii* without resistant genes vs. nonbacterial infection			
Ribosome	67.25	0.00	69
COVID-19	51.62	0.00	77
Pentose phosphate pathway	6.31	0.00	9
Salmonella infection	5.46	0.00	38
Spliceosome	4.43	0.01	24
Sulfur metabolism	4.40	0.01	4
Actin cytoskeleton regulation	4.24	0.01	32
Shigellosis	4.02	0.02	35
Pathogenic Escherichia coli infection	3.99	0.02	29
Viral myocarditis	3.81	0.02	11

NDM, New Delhi metallo-β-lactamase; COVID-19, coronavirus disease 2019; KEGG, Kyoto Encyclopedia of Genes and Genomes.

**Table 3 biomedicines-13-00142-t003:** Significantly expressed KEGG pathways (immune system, signal transduction, signaling molecules and interaction, and cellular processes) in the *Acinetobacter baumannii* without resistant gene infection-associated pulmonary host response.

Subclass (Class) of the KEGG Pathway	Description	Enrichment Score	*p*-Value	Genes in the List
*Acinetobacter baumannii* with NDM infection pulmonary microenvironment				
Immune system (Organismal systems)	Thyroid hormone signaling pathway	9.78	<0.01	42
	Parathyroid hormone synthesis, secretion, and action	4.59	0.01	31
	Glucagon signaling pathway	3.89	0.02	30
	Insulin signaling pathway	3.45	0.03	36
	Cortisol synthesis and secretion	3.25	0.04	19
Signal transduction ((Environmental information processing)	Endocytosis	4.90	0.01	65
	ErbB signaling pathway	4.65	0.01	26
	AMPK signaling pathway	3.76	0.02	33
	Actin cytoskeleton regulation	3.45	0.03	54
Signaling molecules and their interaction ((Environmental information processing)	HIF-1 signaling pathway	3.14	0.04	29
Cell growth and death ((Cellular processes)	Mitophagy-animal	3.01	0.05	20
*Acinetobacter baumannii* without resistant gene infection in the pulmonary microenvironment				
Transport and catabolism (Cellular processes)	Phagosome	3.33	0.04	22
Cell growth and death (Cellular processes)	Ferroptosis	3.11	0.04	8
Cell motility (Cellular processes)	Actin cytoskeleton regulation	4.24	0.01	32

NDM, New Delhi metallo-β-lactamase; KEGG, Kyoto Encyclopedia of Genes and Genomes.

## Data Availability

Data are not available because of privacy or ethical restrictions.

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
