# Peer review of "Investigating the Pulmonary Host Response of Acinetobacter baumannii Infection-Associated Pneumonia by Metagenomic Next-Generation Sequencing"

_biomedicines, 2025, doi:10.3390/biomedicines13010142_

Round 1

Reviewer 1 Report (Previous Reviewer 1)

Comments and Suggestions for Authors

The authors made the suggested modifications, which improved the presentation of the work.

Reviewer 2 Report (Previous Reviewer 2)

Comments and Suggestions for Authors

This manuscript now modified appropriately.

This manuscript is a resubmission of an earlier submission. The following is a list of the peer review reports and author responses from that submission.

Round 1

Reviewer 1 Report

Comments and Suggestions for Authors

The manuscript “Investigating the pulmonary host response of Acinetobacter baumannii infection-associated pneumonia by metagenomic next-generation sequencing” addresses a relevant and pertinent topic in its field of study. The methodology used by the authors is appropriate, the results obtained are very interesting and are presented clearly. However, the authors acknowledge that the study has two very relevant limitations:

1.      An important limitation of the study is the sample size, which suggests the need for a larger number of samples for results that provide greater support for the conclusions..

2.      The authors report that the mNGS results for the three patients detected other potential pathogens, indicating that the infections were not exclusively caused by A. baumannii.  This is an important limitation of the study, so I suggest that the format be a communication, not an Article

Minor corrections

Abstract

Line 18 Write complete Acinetobacter

Line 23 define NMD

Discussion

Lines 386-389 review the text, it seems that there are words that correspond to the Biomedicines format

Conclusions

Line 392 The first statement is true; However, it has nothing to do with the objective of the study, so I suggest eliminating it from the conclusions.

Figure 5 should not be presented in conclusions. I suggest that Figure 5 be explained and placed in Discussion as it constitutes a summary of the findings

Reviewer 2 Report

Comments and Suggestions for Authors

The manuscript “Investigating the Pulmonary Host Response of Acinetobacter baumannii Infection-Associated Pneumonia by Metagenomic Next-Generation Sequencing” is to provide the Host Response by A. baumannii infection.

At line 18, 21, 23, 26, 29, 35, and so on, A. baumannii must be italicized.

It is not clear what authors want to say in “In pulmonary host response to pneumonia caused by A. baumannii with NDM, five immune system-related pathways were identified. Five pathways related to signal transduction were identified. No significant differences were observed in the immune system and signal transduction pathways in pulmonary host response to pneumonia caused by A. baumannii without resistant gene.” First 2 sentence meaning are redundant, but more interestingly third sentence is totally reversed meaning.

Unfortunately, it is not clear what authors want to say in abstract results section.

At line 44 Authors need to add reference for the sentence “Significantly, the frequency of multidrug-resistant (MDR) gram-negative (GN) bacteria as the bacteriologic cause of HABP/VABP is …”

At some point in the sentence from line 52, authors might want to say the infection mechanism, but it looks like continuation of resistant mechanism, and the meaning of the sentence “In clinical applications, the host genetic sequences are discarded, leaving only the pathogen sequences for identification[8].” is not clear, so authors need to clarify that.

At line 79, A. baumannii with NDM (before treatment) should be a A. baumannii without NDM?

At line 80, What does the group of “A. baumannii without resistant gene” mean?

At line 100, Baumannii must be start with lower case.

It is not clear why authors not include the Ab with NDM (Post treatment) group with others?

At line 110, Candida albicans (C. albicans) must be italicized.

Authors need to add some information for the NDM, because even though authors group the patient based on that, none of information was provided.

It is not clear why authors though NDM is important to divide the group?

Even though authors provide prow chart for the experiments, authors did not provide detailed protocol how to prep the sample.

It is not clear why there is no references in the 2.3. mNGS Process.

I could not find the statistics in this manuscript.

It is not clear what make the difference between table 2 and 3?

If authors analyzed based on NDM, authors need to add some discussion.

Comments on the Quality of English Language

Some pharagraphe were written not clearly.